# Heritage, geographical scale and didactic potentiality: Students and teachers' perspectives

**Ana Isabel Ponce Gea**[1]☯*, **Carlos Martínez Hernández**[2]☯, **María Luisa Rico Gómez**[1]☯

**1** Department of General and Specific Didactics, Faculty of Education, University of Alicante, Alicante, Spain,
**2** Department of Social, Experimental and Mathematics Sciences, Faculty of Education, University of Complutense of Madrid, Madrid, Spain

☯ These authors contributed equally to this work.
\* anaisabel.ponce@ua.es

**Data Availability Statement:** Anonymized dataset is available in this research (S1 Dataset).

**Funding:** The authors received no specific funding for this work.

## Abstract

Heritage and space establish reciprocal relations that have been studied for decades. On the one hand, heritage has been described as an inherently spatial phenomenon. On the other hand, places are defined according to the attributes that make up their identity, among which heritage is a fundamental instrument. On the basis of the idea that education plays an important role in the socialization process, transmitted by the inherited culture, to integrate each subject within the specific community, and the notion of scale as the closest to heritage, we defined as general objectives to determine the relationships between geographic scales, heritage perspective and the didactic potential granted to heritage, within the framework of the construction of collective identities, and to contrast the perspectives of students and teachers regarding the geographical scale, heritage and their didactic potential, deducing implications for educational practices. In order to answer to these objectives, we carried out a non-experimental quantitative research, with a relational-predictive objective. Specifically, we used a survey method, being the context the whole of the local scale (Fuente Álamo, Murcia, Spain) and acting as participants all students and teachers of Secondary Education (n = 459) linked to social sciences. They answered the Test on Didactic Potentiality of Heritage according to Scale (TDPHS), and its information was analysed through different procedures (Spearman's correlations, descriptive statistics, Mann-Whitney U. . .), using the statistical programs SPSS. The results show, on the one hand, that the scalar perspective scores are generally low, heritage perspective is consistent with the consideration of the scales, and the perceived didactic potential in relation to heritage is related to the importance given to each of the scales; and, on the other hand, the contrast in the perspectives of students and teachers regarding the geographical scale, heritage and their didactic potential is minimal.

**Competing interests:** The authors have declared that no competing interests exist.

## Introduction

The study of the relationships between spatial and temporal dimensions is nothing new. Neither is it from the educational field, where space and time can be considered as two fundamental dimensions, both conceptually and organizationally, which also act in an interrelated way.

Heritage, as a trace of the past, cannot ignore the spatial dimension, adding the particularities of its conceptualization to the relationship with the space that establishes the historical dimension.

Graham, Ashworth and Tunbridge [1], on the one hand, characterize heritage as an inherently spatial phenomenon. All heritage occurs somewhere, and it is in that place—and not another—for some reason. Thus, heritage is distributed unevenly throughout the space, causing the existence of places with a greater or lesser amount of cultural heritage, but, in any case —and of a greater importance—, different in each case. However, the connection heritage object and place cannot be understood as a 1–1 relationship, coming the concept of scale into play, an intrinsic attribute of existing places through a hierarchy. In the words of same authors, the notion of scale complicates the conceptualization of heritage, since each heritage element belongs to several scales and in each one of them receives an interpretation that can be conflicting [2]. In other words, each heritage element is located in a place, conceptualized from different perspectives that arise from different geographical scales. These are understood as spatial configurations where sets of practices and, therefore, symbolisms [3], product of a social contextual and temporal construction, are included. They are based on different cells of social interaction (family, school, politics, etc.), which are giving new resignifications.

On the other hand, places are defined according to the attributes that make up their identity, what we could call territorial identity [4]. When dealing with our identities, it is difficult to know well where they begin and end and, above all, what are the essential factors that make them up [5]. What is certain is that identity is associated with the similarity and awareness of belonging with respect to a community and the difference regarding the "others". In this process of definition and differentiation, the construction of an identity is associated with the collective memory of a community, recognizing heritage as one of the main memory infrastructures [6] and, at the same time, as indispensable for the construction of territoriality [7]. In addition, a reconciliation between elements of a particular past with those of the community, based, specifically, on a symbolic universe linked to places [8] is necessary to reinforce the link between individual and social identity. These have a duration reflected in tangible or intangible objects, on which identity and the feeling of attachment are configured, supported by the look that is taken to it as a progression, within a social framework of memory [9]. However, just as we mentioned, a heritage element belongs to and can be interpreted from different scales, in the construction of identity there is also the idea of identification scales of a group, linked, in part, to the geographical dimension. There is no single social memory, but as many as diverse interests, conflicts and values take place, the result of particular contexts [10, 11]. The line of separation between the different scales of memory (individual, collective, cultural) is not clear, whereas the social frameworks in which their plurals are also inserted: family, location, State, nation, etc. Thus, each subject is traversed by multiple identities and collective memories that intertwine and overlap each other [12], frequently associated with physical spaces that serve as a framework for argumentation.

Therefore, the interest of reflection and research is not limited to the study of heritage-scale relationships but consists in specifying the true value of scale in the conceptualization and heritage perspective and, above all, in seeing how scale and heritage work together [3] for the construction of identities. Do we have the same contextual tools to define heritage at all scales? Is the emotion, admiration or identification greater for those elements that are closer to us? Or

has the idea of globality broken the Piagetian line that goes from near to far, also diluting the construction of collective identity? Or even more so, are we global citizens or patriots according to what?

Before finally specifying the question in the educational field, we consider it interesting to dwell on some ideas about scale, heritage and identity.

First, the conception of the global world, with continuous exchange of goods, people and ideas [13], involves a continuous reformulation of the spatial dimension, so that what is distant is not so far, nor the near is so exclusively ours. Conceptions continually change, under the paradigm of liquidity [14], and the geographical scales take on another meaning, as the geographical discipline and its incursion into our schools must take on another meaning [15].

As an abstraction, spatial scales blur and overlap, in a non-correspondence with political borders, gaining importance the sense of belonging and social construction [16] at the same time making it difficult to define scale. Thus, the scale is seen as a space under construction [17], conditioned by the social interactions of a community. This space implies the idea of appropriation, geographically delimited with a physical and symbolic identity value, which contains an experiential, existential, material and immaterial dimension, of local social, historical and cultural relations, which share common codes. The territory has a historical cultural continuity, in which the old interacts with the new, in a process of re-functionalization [18, 19]. This sense of territoriality, apart from being configured by the succession in time of cultural guidelines and assets, can also be generated by the external action of higher institutions to support it and represent the community. Or as an interrelation between the factors of the local, endogenous culture, and the guidelines established by the external culture, whose confluence determines an identity associated with a specific space and time [20]. In any case, it is not only linked to a construction in a space, but to the social and cultural differences associated with a territory, at different scales, as open dynamics, within a process of transformation and interaction at a local and global level [21], that can never be considered complete.

Second, in this endowment of significance to define the identity of a community, the heritage not only refers to tangible elements delimited in a physical area, but also to a broad set of values and historical, economic, environmental, and cultural meanings, within the contexts in which they are configured [22]. In fact, there is a process of reconfiguration and spatial self-representation in the assignment of meanings and values, between past, present and future, beyond the conservation of the object itself.

Despite globalization and the configuration of a global identity—which, in heritage terms, makes us consumers [23, 24] of museums, monuments, etc., in a physical and / or virtual journey around the world—, for the most part, heritage clings to the local dimension, to local empowerment, where a community recognizes and values an element -tangible or intangible-because it represents its history and identity, as a place where the memory is stored to make people remember [24]. Heritage is, therefore, a human construction associated with the concepts of identity and belonging, a feeling of being, which affects the agents of a community [25]. From this vision of heritage as inherited is the need for its conservation explained, since it is something built and owned, of which there is awareness of belonging and worth [26]. This motivation of historical argumentation is associated with the most traditional perspective applied to heritage, the commemorative [27], which, while recognizing the role of heritage in shaping collective identities, is frequently related to a single narrative and a hard, objective identity defined as closed [12]. However, heritage must be understood in a process of construction and reconstruction, not static, from contemporaneity with new meanings [28, 29]. Indeed, every transmission process entails reinterpreting the past giving it new meanings from the present, in a retrospective and prospective sense, based on interests to legitimize a collective identity, because memory preserves from the past what is capable of being alive in the present

of a group defined on a territorial scale [30]. Thus, heritage is presented as useful, not only from its economic profitability, but also as a historical source for the construction and reconstruction of the various existing memories around the same heritage element [31, 32]. This utility perspective makes even more sense if we pay attention to the spatial, considering, on the one hand, the diverse interpretations to which a heritage element is exposed according to the scale; and, on the other hand, the multiple, hybrid identities in permanent transformation [12, 28, 29] resulting from the exchange of people and ideas throughout the world.

Against this background, education plays an important role in that socialization process transmitted by the inherited culture to integrate each subject within the specific community, as a process of identification. It is a learning that requires the transmission of new but also present cultural elements in the collective imagination of belonging, which it shares and transfers to future generations [33]. Through this transmitted cultural legacy, which has its historicity, patterns of behaviour, conceptions, etc. are acquired. In other words, an identity supported by traditions retained in memory and materialized in cultural heritage, tangible or intangible, which shows a temporality [34].

Heritage awareness, supported by knowledge and appreciation of heritage, allows its transmission, enjoyment and its care [35]. Education has an important weight to value heritage assets as elements of integrating identity, always accepting and respecting those of other cultures: it helps to develop an awareness of participation in respect, interpretation and transmission of that legacy, as an attitude of civility and social integration, as well as interpretive recognition of space and time typical of a common memory. But always from the clear conception that the memory transmitted as shared by a group of citizens and represented in the form of inherited heritage as a social need [36], is the result of an imaginary and symbolic construction and deconstruction process of meanings based on the associated historical narrative and context.

Therefore, given the "conflictive" idea of heritage, of what elements are necessary to be declared as heritage in each context and moment, due to the significance given to its value in a certain group, it is necessary to know not only the heritage perspective, but also the scalar perspective as an option of the spatial more in line with the heritage, and how these work together in the configuration of identity. The above, in the educational field, presents a new dimension in relation to the didactic potentiality, that is, what heritage elements of each scale deserve to be incorporated into teaching-learning situations. Hence the need for a study on the perception of students and teachers regarding the geographical scale, heritage and the didactic potential of the above for the construction of identity. The convergence or divergence between the teaching and student perspectives will define classroom practices and, above all, the relationships between heritage-scale that are disseminated as part of the collective memory.

## Objectives

Based on the above, we define two general objectives for this research:

G.O.1: To determine the relationships between geographic scales, heritage perspective and the didactic potential granted to heritage, within the framework of the construction of collective identities.

G.O.2: To contrast the perspectives of students and teachers regarding the geographical scale, heritage and their didactic potential, deducing implications for educational practices.

For its resolution, we propose to respond to the following specific objectives:

S.O.1.: To describe the perspectives for each of the geographic scales, attending to variables related to their knowledge, interest, concern, degree of identification and defence.

S.O.2.: To delimit the heritage perspective that underlies the responses of the participants, in accordance with the characterization of the heritage elements offered.

S.O.3.: To determine the didactic potential granted to heritage, establishing relationships with the heritage perspective and geographical scales.

## Method

### Context and participants

To carry out the study, we considered a municipality in the south-east of Spain with about 16000 inhabitants as a context. This choice came from the decision to limit the geographical scale variable to a single case, considering that the uniqueness of each territory is influential in the perspective on the geographical scale and its heritage elements. Thus, in our research, the scales correspond to the municipality of Fuente Álamo (local scale), Autonomous Community of the Region of Murcia (regional scale), State of Spain (national scale) and the world (global scale). At the local scale and for one of the variables included in the instrument, the differentiation between the town of Fuente Álamo, municipal capital and place of educational centres and other infra-municipal entities (districts) was considered.

The initial participants of the research (n = 512) corresponded to all the students of the compulsory secondary education centres of the municipality (n = 506), as well as the social sciences teachers (n = 6) of the same students. We are referring, then, to an intentional non-probabilistic sample [37], considering all the students in the centre and all the teachers who taught the same students and who were more closely related to the issue of heritage from the point of view of this work (history and geography teachers). In the analysis of the information, it was considered the elimination of all participants who had not answered to the entire test, 53 students and no teachers, which meant a reduction of the final sample (n = 459), 453 students and 6 teachers, with 225 men and 234 women. This sample corresponds to the two public centres and the subsidized centre existing in the municipality, being 98.7% students and 1.3% teachers. The age of the participants ranges between 11 and 20 years old (M = 14.84, SD = 1.82), in the case of students, and between 20 and 50 years (M = 37.83, SD = 9.75), in the case of teachers; of which 69.9% are residents of the municipal capital and 30.1% of their districts.

### Research design and information collection: Instrument and procedure

The study carried out responds to a non-experimental quantitative research, with a relational-predictive objective. Specifically, we use a survey method, within the framework of a controlled investigation [38]. This choice responds to the need to cover the whole of a context, involving all the agents and, at the same time, delimited by the local scale chosen for the study, which also conditioned the type of sample.

Once a review of previous information collection instruments was carried out, and in the absence of a test that covered our purposes, an information collection instrument was designed for the specific objective of this study. The Test on Didactic Potentiality of Heritage according to Scale (TDPHS) (S1 Appendix) was built around three theoretical pillars that condition the dimensions and blocks of the test (scalar perspective, heritage perspective and didactic potentiality of heritage), to which a first block of personal information is added. We briefly describe the test configuration.

BLOCK 1, Personal information, is limited to the inclusion of variables that allow us to characterize the sample, as well as groups of interest for the analysis. Specifically, the variables

of age, gender, year, nature of the centre (public, private or subsidized) and place of origin are included.

BLOCK 2, Scalar perspective, consists of a single test question, with five Likert-type items, from (1) Strongly disagree to (4) totally agree. To do this, it is requested to show the degree of agreement or disagreement in relation to the interest, concern, identification, defence and knowledge for each of the four geographical scales (local, regional, national and global), represented in the personalized territories with scaling progression starting from their own municipality.

BLOCK 3, Heritage perspective, consists of two questions that share the objective of knowing the heritage perspective, attending both the epistemological and emotional point of view, and opting for the selection of qualifiers corresponding to the three delimited categories ("Heritage-object", "Heritage-commemoration" and "Heritage-resource") as a response method. The first question arises from the generality of the heritage concept, offering three answer options (a) old, beautiful, strange; b) our, own; c) useful, socio-economically profitable, among which the participant has to choose one. In the second question, the participant is asked to select qualifying terms for each of the scales. Due to the typological breadth of the heritage, the item has been repeated for three heritage elements belonging to different typologies: monuments, landscapes and festivities. A final consideration lies in the concrete exemplification of each scale, to move away the response of the students from the gnoseological diffusion of the generic matters. Therefore, the participant does not describe the geographical scale but a heritage element of this scale. It must be considered, therefore, that, depending on the municipality where the test is to be applied, these examples must change and be personalized. This is not, from our perspective, a weakness of the questionnaire but a need protected by the very nature of the heritage element, where the emotional is linked to the relationships established between subject-object, an object that is necessarily modified on the local scale. The original version of the TDPHS that we used for this study is based on the heritage legacy of the municipality of Fuente Álamo, in the Region of Murcia (Spain), exemplifying the remaining scales according to this progression. In any case, the chosen examples respond to highly representative elements, of recognized value and theoretical and social appreciation, to avoid possible ambiguities and / or detachments.

Finally, BLOCK 4, Didactic potentiality, is made up of a single question, aimed at getting to know the perceptions about the didactic potential of heritage according to the geographical scale. For this, a Likert-type scale has been used again, with (1) not of interest and (4) very interesting, the items corresponding to examples of heritage elements of all scales, shown in random order. Taking into account the scale defined for the score and the heritage element, the participants have to select the degree of interest that their inclusion in the classroom teaching proposals provokes, from the point of view of the students and that of the teachers. As with the examples in Block 3, the heritage elements of this block, in addition to corresponding to the different scales, have been designed according to their representativeness and their social and historical value.

The final design described for the TDPHS has been the result of a content validation process, where internal reviews have been combined with the participation of seven external judges, from the field of social sciences and its didactics (n = 3), teachers (n = 2) and agents of the heritage field (n = 2). These experts have evaluated the test according to its adequacy, relevance, relevance and linguistic correctness, in relation to different aspects of the test: presentation, design, content, items and global assessment. The average of their responses, on a scale of five, has been outstanding both by judges and by items (M = 4.75, SD = 0.5). In addition, for the study of agreement between judges, we applied Kendall's W test (W = .733, Chi2 = 22.000, gl = 6) which reveals that there is statistically significant agreement (p = .001) between the

mean traits assigned by the judges (J1 = 1.00, J2 = 2.70, J3 = 5.10, J4 = 5.50, J5 = 4.30, J6 = 5.00, J7 = 4.40), with a high intensity of agreement. In addition, studying the reliability of the test as internal consistency, the Cronbach's Alpha test ($\alpha$ = .907) shows a reliability between high and very high [37].

The TDPHS was applied collectively, according to the groups previously established in the educational centre, with a range of between 20 and 30 male female students. The information collection was carried out in a single phase, applying the test in all groups in a space of two months.

This information collection was carried out, in all cases, by the researcher responsible for the research, using the reference classroom of each group as space and always within school hours (one session was used from the Geography and History area). Although the instructions were included in writing in the test, the basic instructions for their completion were orally addressed, also resolving doubts during its session. The participants had a maximum time of 50 minutes, having to write down their answers in the test booklet itself.

This research has been developed under a collaboration agreement between the Regional Office for Education, Youth and Sports of the Autonomous Community of the Region of Murcia and the University of Murcia, also counting on the favourable report of the Research Ethics Committee of the academic institution. Likewise, we have the informed consent of the participants involved, ensuring the confidentiality of the results. Anonymized dataset is available in this research (S1 Dataset).

### Information analysis

The data analysis was carried out through four consecutive steps. First, descriptive statistics (means and standard deviations) and Spearman's correlations were calculated for the questionnaire scale. Second, a transformation and recoding of the initial variables was carried out in order to manage the data and the specific objectives set. Third, descriptive statistics were calculated for them. Finally, the relationships between the variables were analysed (Spearman's correlation coefficient) and the defined groups were compared (Mann-Whitney U), taking into account the ordinal nature of the variables and the non-normality of the sample. To carry out these analyses, the statistical program SPSS, version 24, was used.

In Table 1, we include the analysis variables involved, grouped in dimensions.

## Results

### Perspectives for each of the geographic scales

In relation to the first specific objective defined, the variables related to the dimension "Scalar perspective" have been taken into consideration. In Fig 1, the means of each variable are collected, following a progression from lower to higher amplitude of the scale.

As can be seen in the Figure, the interest is higher for the national scale (M = 3.09, SD = 0.89), although closely followed by the global scale (M = 3.01, SD = 1.06). The lowest mean score, in the case of interest, corresponds to the local scale (M = 2.68, SD = 1.03). In relation to identification, the highest marks are found on the national scale (M = 2.66, SD = 1.06) and the smallest on the global scale (M = 2.41, SD = 1.14). Despite the relatively low marks in terms of interest and identification for the local scale, both for concern (M = 2.92, SD = 1.09), as for defence (M = 3.00, SD = 1.09) and knowledge (M = 3.27, SD = 0.99) is the scale with the highest marks; being the one with the lowest marks, in the three cases, the one corresponding to the global scale. According to the scale used for grading, except for three variables that exceed the arithmetic mean of 3.00, the rest are in the range from "somewhat agree" to "quite agree", which rules out the high frequency of close responses to the ideal.

**Table 1. Student variables: Description and measurement.**

| Scalar perspective | | |
|---|---|---|
| **Variable** | **Description** | **Measurement** |
| Interest in scale (local / regional / national / global) | Defined for each of the geographic scales, interest in the scale is understood as the degree of affection, sympathy or affinity that the participant feels for a certain scale. | Likert Scale (1–4) |
| Identification with the scale (local / regional / national / global) | Defined for each of the geographic scales, identification with the scale is understood as the degree of representativeness of the scale for the participant, associated with their identity. | Likert Scale (1–4) |
| Concern about scale (local / regional / national / global) | Defined for each of the geographic scales, concern for the scale is understood as the degree of attention or interest in the scale's problems, as well as their prevention. | Likert Scale (1–4) |
| Scale defence (local / regional / national / global) | Defined for each of the geographic scales, scale defence is understood as the degree of action before the problems of the scale or the need to protect the singularity of this, from an active approach. | Likert Scale (1–4) |
| Knowledge of the scale (local / regional / national / global) | Defined for each of the geographic scales, the knowledge of the scale is understood as the degree of conceptual and experiential management of the participant before each scale. | Likert Scale (1–4) |
| Scalar consideration (local / regional / national / global) | Defined for each of the geographic scales, scalar consideration is understood as the degree of interest, identification, concern, defence and knowledge of each of the scales. | Calculated from the arithmetic mean of the rest of the variables of the dimension. |
| Heritage perspective | | |
| Variable | Description | Measurement |
| Heritage perspective | The variable is defined as the approach adapted to heritage, in relation to its social role, in the configuration of identities and education. | Multiple choice items, using qualifying adjectives. Three categories are established: heritage-object, heritage-commemoration and heritage-resource. In the case of the study by scales, each category is considered a variable marked from 0 to 6, its denominations being of the type "Heritage-commemoration perspective (local)". |
| Didactic potentiality | | |
| Variable | Description | Measurement |
| Didactic potentiality (local / regional / national / global) | Defined for each of the geographic scales, the didactic potentiality is understood as the degree of interest that the heritage element of each scale causes to be included in the didactic proposals of the classroom. | Likert Scale (1–4) |
| Didactic potentiality | the didactic potentiality is understood as the degree of interest that the heritage causes to be included in the didactic proposals of the classroom. | Calculated from the arithmetic mean of the rest of the variables of the dimension. |

From the most global analysis of these variables, we can make some useful appraisals. On the one hand, considering the previous elements for the set of scales, the aspect with the highest marks is interest (M = 2.90, SD = 0.62), followed by defence (M = 2.85, SD = 0.79), concern (M = 2.81, SD = 0.80), knowledge (M = 2.74, SD = 0.66) and, lastly, identification with the scale (M = 2.52, SD = 0.74). In addition, according to the purpose of the work, we study the relationships between the knowledge of the scales and the rest of these variables, which allows us to indicate positive and statistically significant correlations with the interest ($r_{S} = .291$, p < .001), identification ($r_{S} = .315$, p < .001), concern ($r_{S} = .324$, p < .001) and defence ($r_{S} = .353$, p < .001) of the scale.

If we consider the arithmetic mean of all these aspects for each of the scales, the scale with the highest score is the local scale (M = 2.86, SD = 0.77), followed by the national (M = 2.84, SD = 0.73), regional (M = 2.78, SD = 0.71) and the global (M = 2.58, SD = 0.80); none of them exceeding the score of 3.00, corresponding to "quite agree". These markings, beyond the differences that we indicated previously between the different variables considered, allow us to

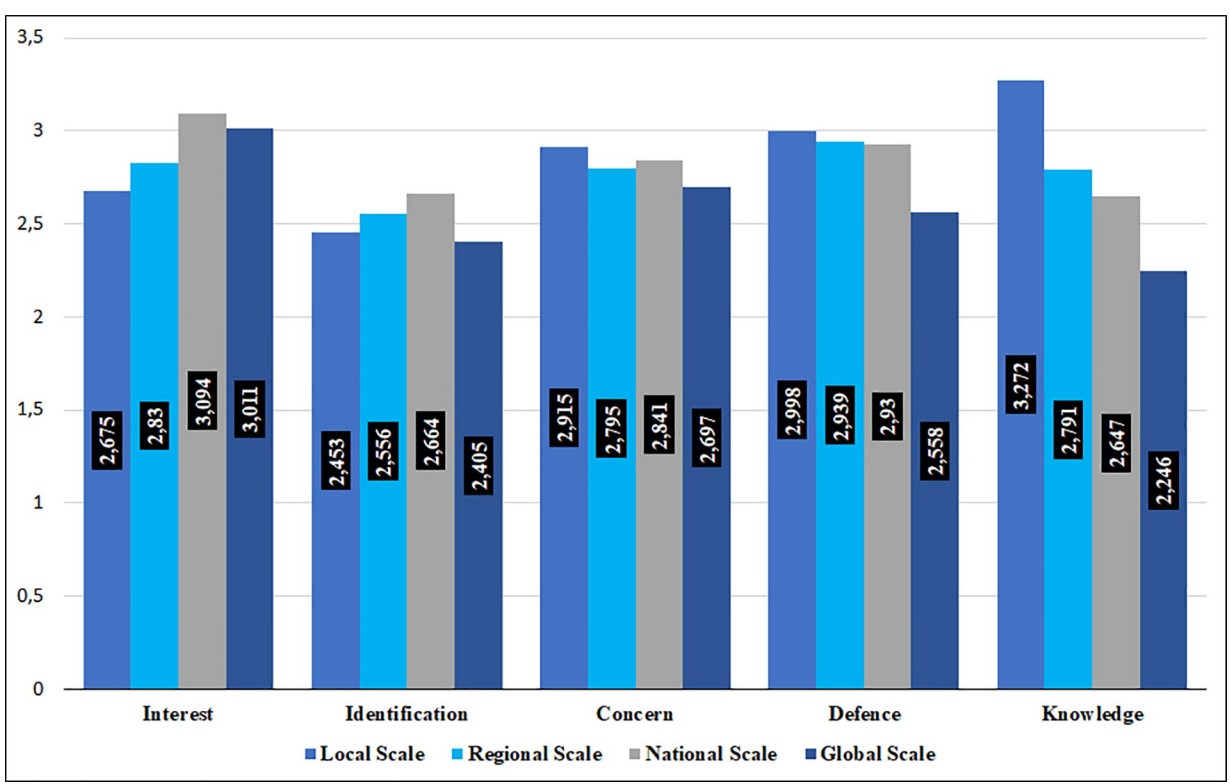

**Fig 1. Perspective on geographic scales in relation to interest, identification, concern, defence and knowledge.**

establish an order in the importance given to the geographical scales, where the municipality is first, followed by the country, the region and last, the global scale.

Taking into account the objective of the comparison between the perspectives of students and teachers, the average marks for all geographical scales are higher in the case of the latter: local scale (student: M = 2.85, SD = 0.77; teacher: M = 3.70, SD = 0.33), regional scale (student: M = 2.77, SD = 0.71; teacher: M = 3.60, SD = 0.46), national scale (student: M = 2.83, SD = 0.73; teacher: M = 3.60, SD = 0.52) and global scale (student: M = 2.58, SD = 0.80; teacher: M = 3.20, SD = 0.77). As can be seen, the order in the consideration of the scales is respected—with a tie in the marks of the regional and national scales, in the case of teachers—and, while the students' scores are between "somewhat in agreement" and "quite agree", the arithmetic means of the teachers, for all scales, are located between "quite agree" and "totally agree".

Applying the Mann-Whitney U test, for two independent samples, being the grouping variable "Profile" (student, teacher), we accept that there is a statistically significant difference between students and teachers for "Consideration of the local scale" (z = - 2.817, p = .005), "Consideration of the regional scale" (z = -2.871, p = .004) and "Consideration of the national scale" (z = -2.707, p = .005). There is, therefore, a difference between the medians of the variables, always higher in the case of teachers. Despite its closeness to statistical significance, the data do not show statistically significant differences for the responses given by students and teachers regarding "Consideration of the global scale" (z = -1.909, p = .056). Analysing in a concrete way the variables involved, there are statistically significant differences between the perspectives of the students and the teachers for the interest (z = -3.260, p = .001) and concern (z = -2.664, p = .008), in the case of the local scale; for interest (z = -2.950, p = .003), concern (z = -2.514, p = .012) and defence (z = -2.221, p = .026), in the case of the regional scale; for

identification (z = -2.764, p = .006), in the case of the national scale; and for knowledge (z = -2.887, p = .004), relative to the global scale.

## Heritage perspective

For the analysis of this objective, the dimension "Heritage perspective" has been taken into consideration, considering the variables, initially, from a global point of view and, later, from the application to the different geographical scales.

Based on the definition that the participants make of the concept of heritage, 48.6% of the responses are associated with the heritage-commemoration perspective, defining it as our own. It is followed in frequency by the heritage-object perspective (38.8%) and, lastly, is the heritage-resource perspective (12.6%).

For the sake of comparing the groups "students" and "teachers", we applied the Mann-Whitney U test, for two independent samples (z = -0.544, p = .586), accepting the null hypothesis of no difference between the medians of "Heritage perspective" according to "Profile". From the above, a similar heritage perspective is derived for teachers and students, coinciding in a prevalence of the heritage perspective of commemoration.

In Fig 2, the arithmetic means relative to each of the equity perspectives according to the scale are shown. These results must be interpreted according to the scale established in the instrument, with scores between 0 and 6.

As can be seen, the heritage-object perspective prevails on the regional (M = 3.66, SD = 1.58), national (M = 3.55, SD = 1.69) and global (M = 4.03, SD = 1.68) scales, while on the local scale, the heritage-commemoration perspective obtained the highest score (M = 3.04, SD = 1.60). Therefore, and in comparison, with the characterization of the generalized heritage concept, the adjectivisation of specific heritage elements associated with a geographical scale finds specific nuances to the consideration of this particular scale. This question is consistent with the correlation study carried out between the variables, which offers positive and statistically significant correlations between the consideration of each of the geographical scales and the heritage-commemoration perspective, not occurring for the two remaining heritage

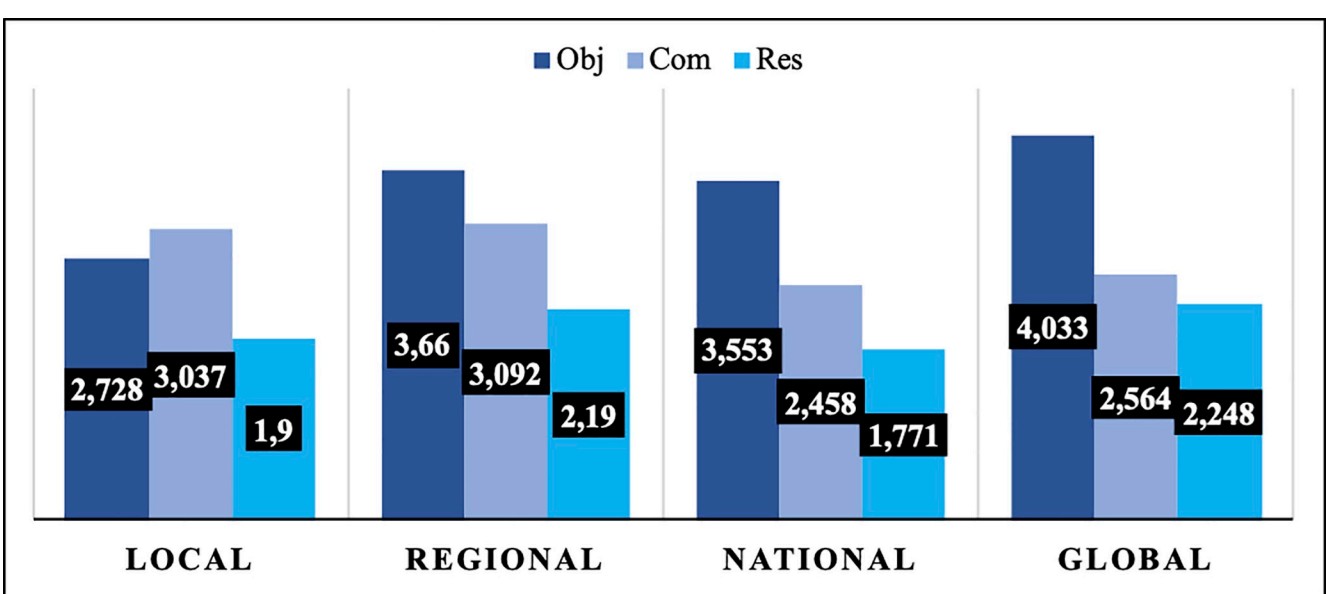

**Fig 2. Arithmetic means of the equity perspective according to scale.**

perspectives. The foregoing takes place, therefore, for "Consideration of the local scale" and "Heritage-commemoration perspective (local)" ($r_{S =}$ .173, p< .001), "Consideration of the regional scale" and "Heritage-commemoration perspective (regional)" ($r_{S =}$ .137, p = .003), "Consideration of the national scale" and "Heritage-commemoration perspective (national)" ($r_{S =}$ .116, p = .013) and "Consideration of the global scale" and "Heritage-commemoration perspective (global)" ($r_{S =}$ .205, p < .001). Despite the weak correlations, a higher consideration of the scale seems to be related to higher scores from the heritage-commemoration perspective versus the heritage-object or heritage-resource perspective. In the same vein, after studying the correlations between the generalized heritage conceptualization and the perspective associated with a geographical scale, the results do not offer positive and significant correlations in any case or for any heritage perspective, except for a single positive and significant correlation, albeit weak, between "Heritage perspective" and "Heritage-resource perspective (local) ($r_{S =}$ .107, p = .022).

Considering the comparison between groups, we include in the Table 2, the descriptive statistics for each of the variables involved in this second specific objective, depending on the group "Students" or "Teachers".

Considering the results of the previous Table, reflections can be drawn between the groups of participants beyond the global arithmetic means. Students and teachers agree on the most frequent heritage perspective for the local scale and for the global scale. Whereas the participants characterize the heritage elements of the closest scale by means of an identity argument of belonging and ownership associated with a hard identity, both groups coincide in defining the heritage elements of the global scale using qualifiers without historical or identity argumentation, such as antiquity or beauty. In the case of the regional scale and the national scale, the responses follow the same pattern: the highest arithmetic means correspond to the category of heritage-object, for the students, while, in the group of teachers, the highest scores correspond to the heritage-commemoration.

However, after applying the Mann-Whitney U test, for two independent samples, the grouping variable being "Profile" (student, teacher), we conclude that there is no statistically significant difference between the medians of the previous variables, except for "Perspective heritage-object (regional)" (z = -2.215, p = .027), with a higher median in the case of students.

**Table 2. Descriptive statics for heritage perspectives depending of geographic scale and group.**

|  | Students | | Teachers | |
|---|---|---|---|---|
|  | M | SD | M | SD |
| Heritage-object Perspective (local) | 2.74 | 1.40 | 1.50 | 1.76 |
| Heritage-commemoration Perspective (local) | 3.02 | 1.60 | 4.00 | 1.10 |
| Heritage-resource Perspective (local) | 1.91 | 1.42 | 1.33 | 1.97 |
| Heritage-object Perspective(regional) | 3.68 | 1.57 | 2.17 | 1.33 |
| Heritage-commemoration Perspective (regional) | 3.09 | 1.72 | 3.17 | 1.47 |
| Heritage-resource Perspective (regional) | 2.19 | 1.67 | 1.83 | 1.72 |
| Heritage-object (national) Perspective | 3.57 | 1.69 | 2.67 | 1.75 |
| Heritage-commemoration Perspective (national) | 2.45 | 1.58 | 3.00 | 1.41 |
| Heritage-resource Perspective (national) | 1.78 | 1.62 | 1.17 | 1.17 |
| Heritage-object Perspective (global) | 4.05 | 1.67 | 2.83 | 1.72 |
| Heritage-commemoration Perspective (global) | 2.58 | 1.44 | 1.67 | 1.63 |
| Heritage-resource Perspective (global) | 2.26 | 1.59 | 1.50 | 1.22 |

## Didactic potentiality and its relations with the heritage perspective and geographical scale

For the analysis of this third objective, we consider the variable "Didactic potentiality", understood as the interest that the heritage element arouses for its use in the classroom, from a generalized analysis and according to geographical scale.

After studying the didactic potentiality of the set of heritage elements, regardless of their scale, the heritage presents between "somewhat interesting" and "quite interesting" for its incorporation in the classroom (M = 2.41, SD = 0.48), with expected differences between the students (M = 2.40, SD = 0.48) and the teaching staff (M = 2.91, SD = 2.91). Despite being able to accept that there is a statistically significant difference between the medians of the variable "Didactic potentiality" for teachers and students (z = -2.013, p = .044), none of the arithmetic means of the groups reaches level 3, which entails relevant implications, at the didactic level, not achieving a perception of "quite interesting" for any of the groups.

In Table 3, the most relevant results in the analysis of the variable according to the geographical scale and profile of the participant are shown.

According to the above, those heritage elements that cause a greater interest for their incorporation in the classroom are those of the global scale, followed by the regional, national and, finally, the local, both for students and teachers. In fact, among the only four elements that exceed the arithmetic mean of 3.00, two belong to the global scale (Pyramids of Egypt and Great Wall of China), one to the regional scale (Roman Theatre of Cartagena) and one to the national scale (Way of Saint James). None of the heritage elements that are included as part of the local scale exceed the arithmetic mean of 2.50.

Applied the Mann-Whitney U test, in relation to the variables included in the Table 3 and being the grouping variable "Profile" (student, teacher), we accept the alternative hypothesis of statistically significant difference between the medians for the variables "Didactic potentiality (regional)" (z = -2.376, p = .018) and "Didactic potentiality (national)" (z = -2.241, p = .025); not existing for the rest of variables a statistically significant difference depending on the grouping variable. This implies that the degree of interest of heritage elements to be used in the classroom is similar for the local and global scales, in students and teachers.

Applying the Spearman's correlation coefficient between the variables related to the didactic potentiality, the geographical scales and the heritage perspective, we can point out positive and significant correlations between the consideration of each of the scales and the perceived didactic potential for each of these (Table 4).

Despite being, in all cases, low correlations, a greater consideration of the geographical scale implies a greater didactic potential attributed to the heritage elements belonging to it. This type of relationship does not take place, however, between the heritage perspective (heritage-object, heritage-commemoration and heritage-resource) and the didactic potentiality of heritage elements.

**Table 3. Descriptive statistics for didactic potentiality depending on geographic scale and group.**

|  | Didactic potentiality (local) | | Didactic potentiality (regional) | | Didactic potentiality (national) | | Didactic potentiality (global) | |
|---|---|---|---|---|---|---|---|---|
|  | **M** | **SD** | **M** | **SD** | **M** | **SD** | **M** | **SD** |
| Student | 2.14 | 0.68 | 2.55 | 0.63 | 2.27 | 0.61 | 2.67 | 0.57 |
| Teacher | 2.71 | 1.11 | 3.25 | 0.63 | 2.83 | 0.50 | 2.86 | 0.93 |
| Both | 2.15 | 0.69 | 2.56 | 0.64 | 2.28 | 0.62 | 2.67 | 0.57 |

**Table 4. Correlations between the variables related to teaching potential and geographic scale.**

|  | Did. pot. loc. | Did. pot. reg. | Did. pot. nat. | Did. pot. global |
|---|---|---|---|---|
| Local Cons. | .329*** |  |  |  |
| Regional Cons. |  | .336*** |  |  |
| National Cons |  |  | .287*** |  |
| Global Cons. |  |  |  | .319*** |

## Discussion

At the beginning of the study, and in accordance with the theoretical framework, we set ourselves two general research objectives, around which we discuss the results obtained.

On the one hand, we intended to determine the relationships between geographic scales, heritage perspective and the didactic potential granted to heritage, within the framework of the construction of collective identities. In this line, it is important to highlight that the relationship between the social identification of heritage on a geographical scale is conceived as linked to the enhancement of the appropriation of elements that offer cultural sense of community. This feeling of appropriation and enhancement of the cultural heritage is connected, as we referred in the theoretical framework, to the identity process and the different degrees of loyalty to the group, according to the collective characteristics considered as attributable to the subject [39].

With this in mind, the analysis offers results that, in a first reading, might seem contradictory. Nevertheless, we believe they are of interest for reflection on the construction of identities.

First, the scalar perspective scores are generally low, which does not suggest, a priori, a well-defined territorial identity [4]. Among them, the most valued scale is the local scale, followed by national, regional and, ultimately, global. This progression seems to recognize the importance of subjectivities in the appraisal of scales and the construction of identities, leaving in the background those scales that do not directly concern the particularities of its citizens [40]. Between the nation and the person there are other scales that are much more related to the life of each individual, particularly the local one, where, even within individuality, all members of a community remember, think and act through common codes; they have elements of present identification resulting from a living past, retained in the collective memory. Even more complicated is the search for the common ground if one speaks of the world, since it is difficult to maintain a memory that contributes to a world identity, due to the fact of maintaining ties between the individual and becoming global that psychologically unite a common memory that is hold over time. Thus, global citizenship in the interconnected world [5, 14] continues to be more a wish than a reality [41, 42]. Indeed, the closest and most singular identity references that categorize the subject of a broader social group respond to an idea of distinction with respect to the other, as an element of identification based on particular experiences that the other has not lived. It is an identity that is constructed in mental representations from the culture in which it is inserted [43].

All identity is a social product based on mental representations, such as acts of perception, evaluation, knowledge and recognition and their relationship with object representations [44]. Hence, however, this question of the consideration of the local requires a reflection in relation to the most marked aspects: knowledge, concern and defence. As opposed to these elements linked to closeness and, we could say, to the emotional, the national scale prevails when it comes to identification and interest. This, possibly, may indicate that the close, the local, clings to the emotional, as a feeling of belonging, while identification is more closely linked to the socio-political construction of the national dimension. Indeed, the identity of the subject, as

far as the social dimension is concerned, is produced by the internalization of a culture that already exists, is inherited, through contextual milestones imposed by the external factors, at the socio-political level, by superior external instances, or that have been acquired by the group insertion itself [45]. Thus, along these lines, this differentiation does not seem strange, specifically, when the *grands récits*, which have offered a unique and privileged vision of the past [46], were a basic tool for the creation of nations [1], promoting an authoritarian, homogeneous identity and with a sense close to the biological dimension [47]. Thus, this feeling of identification, the result of an identity process from power, has been built through speeches and acts focused on the national, as a political subject [48] forgetting the regional scales—at least, for the part of the country that concerns us–, and differentiating what is "ours" from what is "other", in a distinction that goes beyond the spatial factor [49].

Second, the results regarding the heritage perspective are consistent with the consideration of the scales. The conceptualization of heritage, from a generalist approach, entails a clear prevalence of the commemorative, followed by heritage as an object and, finally, as a resource. Thus, we assume a symbolic-identity perspective [27], compatible with Nietzsche's monumental history, in which heritage would have to be preserved for those who come later. Heritage is symbolic, because it represents; it is emotional, because it is the emotions that give meaning to the subject-object relationship that we mentioned at the beginning; it is ours, because we feel that it belongs to us; and it is territorial, because we define that belonging according to a physical space, despite its abstraction. This generalist vision finds nuances in its reading according to geographical scales. Whereas the local scale is associated with this category of heritage-commemoration—that of heritage as a concept, in the abstract—, the rest of the scales show higher scores for heritage as an object. We return again to the idea of what is close for the explanation of the most emotional heritage perspective, which is also symbolic and reinforces the identity of the closest scale [1]. This association of the local scale with a commemorative perspective is not strange, since the greater the knowledge of the scale, the greater the prevalence of this type of perspective. It is precisely, asked by the local scale, when the participants recognize a highest conceptual and experiential management. Indeed, because of that need of individuals to differentiate from the rest through a daily and experiential cultural landmark. In addition, other studies developed on the conceptualization of heritage in secondary education students, both inside and outside of Spain, coincide in the commemorative perspective as well as that of local heritage [50]. Thus, we could say that the heritage perspective of a certain scale is more related to the scalar perspective than to the heritage perspective in the abstract, which implies that, when a territorial identity is not clearly defined, heritage changes from being a symbol of the community to become an object of consumption without historical or identity argument. In any case, the dissolution of that harder identity—perhaps more patriotic—associated with the heritage perspective of the commemorative level, does not find an alternative to the perspective of heritage as a resource, which would allow its consideration as a source based on which multiple narratives could be built, welcoming diverse perspectives, and configure more inclusive identities [12, 51].

Third, the perceived didactic potential in relation to heritage as a shaping factor of identities yields controversial results from its didactic implications and from the confrontation with the importance given to each of the scales. In general terms, heritage is described with a low didactic potentiality. Despite its eminently interdisciplinary nature, the interest that can be deduced for its incorporation in the didactic proposals of the classrooms is not high. However, there are differentiations according to the scales whose discussion finds connections with the heritage perspective and the configuration of those memories common to the community. In contrast to what happened with the scalar perspective, a greater didactic potential is described for heritage elements on the global scale, followed by those belonging to the regional, national and, lastly, local. Although, at first, it seems contradictory due to the considerations made on each

scale, there are certain factors that could explain, in part, these perceptions. First, the influence of the perspectives of the same subjects on heritage. If we understand heritage as associated with an identity constructed from the symbolic and the emotional, perhaps the association of this type of discourses with the school does not occur, but with other memory spaces where these related narratives are constructed, although they do not necessarily carry enough scientific basis [52]. This positioning would also explain the little recognition of the local, to whose elements they already have access through other memory infrastructures, as does not happen, imperiously, with the rest of the scales. Also, in a misapplication of historical relevance [53], the local scale is likely to provoke the thought that history is taught in school, what important people with names and surnames did and unmade, and not what happens in their town or with its people. Thus, the participants consider that the heritage elements of most interest to the classrooms are those that they treat as objects [54], susceptible to knowledge through anecdote, to which they do not have access in their usual memory spaces and, in any case, distinguished from identity and emotional elements. This would be consistent with the analysis made of the importance given to heritage as an object, monumental and on a large scale. Second, and along the same lines, syllabi do not leave much room for the local level [55], so there is a curricular tradition perceived by students and teachers that invites affinity for other scales compared to the closest one.

On the other hand, the contrast in the perspectives of students and teachers regarding the geographical scale, heritage and their didactic potential could be synthesized by stating that it is minimal. Nevertheless, this consideration, as well as the other results relating to the group of teachers, cannot be assumed to be as robust as those relating to the student group, due to the size of the sample. In relation to the scalar perspective, the scores are higher than in the case of the students, but the order in the scales is, in fact, identical, so that the territorial identity seems shared. Likewise, the heritage perspective associated with heritage as a concept is similar for teachers and students, coinciding with the commemorative aspect that also prevails on all scales except the global one. This issue could be explained by a greater knowledge of the scale on the part of teachers, which makes them get even closer to identity than in the case of students. In addition, the results are in line with other investigations performed [56–58], which reveal a traditional conceptualization of heritage, associated with the monumental, by teachers and on a large scale. Regarding the didactic potential, there are differences, but, in any case, there is no clear interest in its use in the classroom. Perhaps, this positioning is not so much a disinterest as a consideration of the difficulties that are associated [59], largely due to the already mentioned curricular tradition; inclusion, in any case, associated with a heritage perspective that is far from its use as a source.

In summary, given the coincidence between students and teachers around the basic questions of the study and the relationship between the scalar consideration, the heritage perspective and the didactic potentiality, we can delimit two major conclusions, which suggest implications in the field of research and didactics.

The first one has to do with the undoubted influence of other narratives external to the school in the conformation of identities. This agrees with what has been said: that the subject does not have its own and unique identity, but is contextual, temporal and is constituted from different potential and positional dimensions that are not separated and interrelated; without any scale being able to be defined as the only identity and full. Hence the possible contradictions in the major results given at the local, national and global level depending on the geographical perspective, the elements of identification or the didactic potentiality of heritage [60]. The heritage identification of the subject is defined, in this case, at the emotional level that the cultural experience of the singular occurrences involves and by the political imposition of the national dimension. We cannot ignore that the subjects have—we have—an important

background coming from social and family spheres and from the so-called "memory technologies", such as books, films, documentaries, music. . . [61]. These narratives, anchored in the local sphere, are valued by the participants and condition the school message, even reducing their impact on the students to a minimum [62]. It is therefore necessary to work on the reconfiguration of memory narratives in other spaces outside the school, both for students and teachers, from which they could also draw for the constitution of their own perspectives.

The second of them, and the one that concerns us, is related to the need for curricular and conceptual changes in the educational field, both in compulsory education and in the training of future teachers, based on two premises: the need to reconstruct the frameworks of traditional geographical reference points, and the recognition of an unequal emotional charge with respect to heritage elements of different scales, which allows us to establish useful heritage perspectives in accordance with the contexts. On the former, we need to break with the idea of scale as uniformity. Curricular geographic frameworks, with their enormous weight and influence [63], continue to organize around political boundaries and therefore respond to more traditional configurations. The scales are understood as internally uniform and concepts such as those of nation continue to contribute to the differentiation with other people. The movements to which our societies are responding make it urgent to consider the scale as that place where things happen, as a product of social interaction. And also, as realities that are interrelated, overlap, even contradict, in that idea of *glocality* recognizing the uniqueness of the scales, but emphasizing their bidirectional influence. In this sense, if it is possible to teach and learn from the local framework to the national and even international one [24] so that a local product is a source that is contextualized in the different scales, and if it is also the scale most considered by our students, its almost omission in the syllabus, as in the perceptions of our teachers, does not seem the most accurate. On the latter, if we recognize that there are heritage elements with which there is a greater emotional connection and that this fact is related to geographical scales, we should not even condemn certain heritage elements to be located only in memory spaces, in which we do not have influence, nor limit the inclusion of heritage to the heritage-object perspective, which will remain in the sense of the anecdote and the absence of a useful historical argument for the discussion of any type of identity. Thus, the consideration of heritage as a historical source, from which to build multiple narratives, through historical research processes that allow the formulation and reformulation of problems [64, 65], is presented as a good option. On the one hand, because it connects, on the social level, with the possibility of configuring multiple identities based on diverse narratives produced; and, on the other hand, because the historical contextualization that turns the source into evidence [66] cannot take place without considering the temporal, social and, therefore, spatial realities that allow the interpretation of the source.

In conclusion, if the collective identity cannot be defined exclusively through territorial identity, if any strict delimitation of the spatial is obsolete for the construction of identities and if, however, the heritage cannot ignore the influences that the space exerts on its own conceptualization; heritage, identity, emotion and territory must be addressed as a whole to train students for democratic citizenship [67, 68]. Yet, this should not be applied in the search for a new homogeneity, unloaded from the political sphere, which ends up being polarizing; but from a true consideration of the multiplicity of understanding, feeling and valuing the world throughout time and space.

## Supporting information

**S1 Appendix. Test on Didactic Potentiality of Heritage according to Scale (TDPHS).** (PDF)

**S1 Dataset. Data set of the research.**
(SAV)

## Author Contributions

**Conceptualization:** Ana Isabel Ponce Gea, María Luisa Rico Gómez.

**Formal analysis:** Ana Isabel Ponce Gea.

**Investigation:** Carlos Martínez Hernández.

**Methodology:** Ana Isabel Ponce Gea.

**Resources:** Carlos Martínez Hernández.

**Validation:** Ana Isabel Ponce Gea.

**Writing – original draft:** Ana Isabel Ponce Gea, María Luisa Rico Gómez.

**Writing – review & editing:** Ana Isabel Ponce Gea, Carlos Martínez Hernández, María Luisa Rico Gómez.

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
