## [Decision Letter · Decision Letter 0]

8 Apr 2021

PONE-D-21-06778

Heritage, geographical scale and didactic potentiality: students and teachers' perspectives

PLOS ONE

Dear Dr. GEA,

Thank you for submitting your manuscript to PLOS ONE. After careful consideration, we feel that it has merit but does not fully meet PLOS ONE’s publication criteria as it currently stands. Therefore, we invite you to submit a revised version of the manuscript that addresses the points raised during the review process.

All comments must be addressed in detail before re-submission; in particular:

1. Language editing

2. Clarify and extend section on data availability and sample size

We look forward to receiving your revised manuscript.

Kind regards,

Peter F. Biehl, PhD

Academic Editor

PLOS ONE

Additional Editor Comments:

Your manuscript has now been seen by two referees, whose comments are appended below. You will see from these comments that one reviewer has raised concerns that must be addressed in detail before re-submission:

1. Language editing

2. Clarify and extend section on data availability and sample size

Journal Requirements:

4. Please improve statistical reporting and refer to p-values as "p<.001" instead of "p=.000". Our statistical reporting guidelines are available at https://journals.plos.org/plosone/s/submission-guidelines#loc-statistical-reporting

Reviewers' comments:

Reviewer's Responses to Questions

**Comments to the Author**

1. Is the manuscript technically sound, and do the data support the conclusions?

Reviewer #1: Partly

Reviewer #2: Yes

2. Has the statistical analysis been performed appropriately and rigorously? 

Reviewer #1: Yes

Reviewer #2: I Don't Know

3. Have the authors made all data underlying the findings in their manuscript fully available?

Reviewer #1: No

Reviewer #2: Yes

4. Is the manuscript presented in an intelligible fashion and written in standard English?

Reviewer #1: Yes

Reviewer #2: Yes

5. Review Comments to the Author

Reviewer #1: Overall, this is a wide-ranging and intriguing study that covers a lot of ground theoretically. It spans heritage, geography, identity, memory, pedagogy, and sometimes could be more clear and concise in tying these multifarious concepts together in the introduction, but overall does a reasonable job covering this range. It does a much better job connecting theory to method to results. In particular, the goals focused on scale, heritage, and didactic potential are well-addressed very clearly in methods and results. There are some language/editing issues, such as run-on sentences and grammatical errors, especially noticeable within the abstract and in about the first two pages of the manuscript within the introduction. The rest of the manuscript improves greatly in this regard, but perhaps a manuscript editing service is recommendable to address these errors.

Regarding PLOS ONE's data policy, the data does not seem to be included directly here (i.e. the results of each survey); only statistical summaries of the surveys seem to be included in the manuscript. It would of course be reasonable to not include the base data here considering the data is from a vulnerable population (children), but data availability (or lack thereof) should be more explicitly addressed by the authors considering the publisher's policy. Moreover, though it is discussed and summarized, an example of the actual survey instrument is not included; this would greatly improve the clarity of the discussion.

In general, the statistical methodology is broad-ranging and impressive. However, if I am not misunderstanding, while the initial sample size of students at 506 seems satisfactory, there is a sample size of only 6 teachers. From these 6 teachers, many generalized assumptions are made in the results and interpretation comparing students' to teachers' perspectives as a whole. I'm not sure that such broad generalizations can be reasonably made about teachers in general from only 6 subjects. If, perhaps, there are only 6 teachers of relevant disciplines in the study area, this should be made clear and the potential biases of this small sample enumerated by the authors. It is also not clear if any of these 6 were removed from the final analysis, as it is noted that any incomplete tests were not included, with TOTAL n=459 after this exclusion (without clarity about the makeup of students vs teachers in this final sample). This needs clarification beyond a general note that the sample is 'intentional non-probabilistic'.

Overall, this study is recommendable for publication as it makes an interesting contribution to the dialogue on heritage with a unique take integrating education and scale into the discussion with a methodologically rich survey. However, some copyediting issues, data availability issues, and sample size issues should be addressed before publication.

Reviewer #2: The study in the article was presented clearly with terms, research design and execution, and results defined and thoroughly discussed. There are a few minor grammatical errors in the introduction (lines 35, 36, 103) but they do not detract from the clarity of the article. Overall, it presented interesting, original research regarding heritage and provoked questions for further discussion.

6. PLOS authors have the option to publish the peer review history of their article (what does this mean?). If published, this will include your full peer review and any attached files.

Reviewer #1: No

Reviewer #2: No

---

## [Author Response · Author response to Decision Letter 0]

20 Apr 2021

Reviewer 1

Reviewer #1: (…) There are some language/editing issues, such as run-on sentences and grammatical errors, especially noticeable within the abstract and in about the first two pages of the manuscript within the introduction. 

The article has been revised by a professional translator and the changes have been highlighted throughout the paper. In addition, a document signed by the translator on the revision is attached to this resubmission.

Regarding PLOS ONE's data policy, the data does not seem to be included directly here (i.e. the results of each survey); only statistical summaries of the surveys seem to be included in the manuscript. It would of course be reasonable to not include the base data here considering the data is from a vulnerable population (children), but data availability (or lack thereof) should be more explicitly addressed by the authors considering the publisher's policy. Moreover, though it is discussed and summarized, an example of the actual survey instrument is not included; this would greatly improve the clarity of the discussion.

Following the indications of the journal, we indicated the availability of anonymized dataset (p. 13., l. 261) and added it to the resubmission (S2 Dataset). 

In general, the statistical methodology is broad-ranging and impressive. However, if I am not misunderstanding, while the initial sample size of students at 506 seems satisfactory, there is a sample size of only 6 teachers. From these 6 teachers, many generalized assumptions are made in the results and interpretation comparing students' to teachers' perspectives as a whole. I'm not sure that such broad generalizations can be reasonably made about teachers in general from only 6 subjects. If, perhaps, there are only 6 teachers of relevant disciplines in the study area, this should be made clear and the potential biases of this small sample enumerated by the authors. It is also not clear if any of these 6 were removed from the final analysis, as it is noted that any incomplete tests were not included, with TOTAL n= 459 after this exclusion (without clarity about the makeup of students vs teachers in this final sample). This needs clarification beyond a general note that the sample is 'intentional non-probabilistic'.

As the reviewer notes, the initial number of teachers is dependent on the number of teachers in the school in relation to heritage-related areas. This issue is clarified in the paper, as well as the reduction of the sample (p. 9). In addition, we include as part of the discussion the idea that the results cannot be taken with the same robustness in the case of teachers, as the sample is very small (p. 28, l. 530-531).

Reviewer 2

Reviewer #2: The study in the article was presented clearly with terms, research design and execution, and results defined and thoroughly discussed. There are a few minor grammatical errors in the introduction (lines 35, 36, 103) but they do not detract from the clarity of the article. Overall, it presented interesting, original research regarding heritage and provoked questions for further discussion.

As we pointed out, the article has been revised by a professional translator and the changes have been highlighted throughout the paper. Moreover, a document signed by the translator on the revision is attached to this resubmission.

---

## [Editor Report · Decision Letter 1]

26 Apr 2021

Heritage, geographical scale and didactic potentiality: students and teachers' perspectives

PONE-D-21-06778R1

Dear Dr. GEA,

We’re pleased to inform you that your manuscript has been judged scientifically suitable for publication and will be formally accepted for publication once it meets all outstanding technical requirements.

Kind regards,

Peter F. Biehl, PhD

Academic Editor

PLOS ONE
---

## [Editor Report · Acceptance letter]

29 Apr 2021

PONE-D-21-06778R1 

Heritage, geographical scale and didactic potentiality: students and teachers' perspectives 

Dear Dr. Gea:

I'm pleased to inform you that your manuscript has been deemed suitable for publication in PLOS ONE. Congratulations! Your manuscript is now with our production department. 

Kind regards, 

on behalf of

Dr. Peter F. Biehl 

Academic Editor

PLOS ONE